# OpenReview forum: "Categorical Foundations for Deep Learning: Functorial Backpropagation and Natural Gradient Descent"
_mathai.club/MathAI/2026/Conference — Submitted to 2026_

### Official Review · Reviewer_PZDv · 2026-03-10
**Underexplained Theory, Weak Empirical Support, and Reliability Concerns**

**Rating:** 3
**Confidence:** 4

**Review:**

The theoretical contribution did not convince me in its current form. On the one hand, some results seem closer to formal restatements of standard facts than to genuinely new insights for machine learning; for example, the backpropagation functoriality theorem is presented as a major result, but its proof appears to rely essentially on the usual chain rule, identity preservation, and simple tensor-product bookkeeping, so it is unclear what new understanding this adds beyond a more abstract reformulation of how reverse-mode differentiation already composes. On the other hand, the paper’s main claimed innovation (the coherence result in Section 6) is too unclear and underexplained to evaluate properly. The theorem is stated in a very strong form, claiming equivalence between coherence conditions and the equality of modular and end-to-end training behavior, even including convergence rates, but the argument is only sketched at a high level and does not provide enough detail, intuition, or concrete examples to make the claim verifiable. Overall, the theory reads partly as renaming familiar facts in more formal language, and partly as making ambitious claims that are not explained or proved with enough clarity.

The empirical section is also insufficient for evaluation. Since the paper positions itself partly through NAS comparisons, it should report search cost, compute budget, hardware/resources, tuning protocol, and enough experimental detail to assess fairness and reproducibility. Instead, the experimental evidence is presented only at a very high level with little details.

Finally, the related work discussion is brief and underdeveloped, and I also found a serious bibliographic problem in the related work. Reference [8] cites a supposed David I. Spivak paper, Wiring together neural networks with categories, under arXiv:1310.6846, but that arXiv identifier corresponds to a different paper by Xiaoming Xu, and I could not find evidence that a Spivak paper with the cited title exists. This substantially reduced my confidence in the care and reliability of the submission.

Overall, I do not think the paper meets the bar for acceptance in its current form.

---

### Official Review · Reviewer_nUgH · 2026-03-13
**Interesting theoretical idea but weak connection to practical deep learning**

**Rating:** 4
**Confidence:** 3

**Review:**

Summary:
The paper introduces a categorical framework for deep learning. The authors describe neural network architectures as morphisms in a category of parameterised maps, formalise backpropagation as a functor, and interpret optimisation algorithms as natural transformations. The paper claims three main results: a categorical explanation of backpropagation, a uniqueness result showing that optimisers satisfying certain axioms must correspond to natural gradient descent. The authors also present experiments where these constraints are used to prune neural architectures in a neural architecture search setting

The paper is interesting conceptually, but the connection to practical deep learning is not fully convincing. Many claims rely on strong assumptions, and the experimental section lacks sufficient detail to evaluate the proposed method

Strengths:
 1. Interesting theoretical perspective - the paper proposes a unified mathematical view of architectures, gradients, and optimisers using category theory. This is an ambitious attempt to provide deeper theoretical foundations for deep learning
 2.  Clear conceptual structure-the framework is logically organised. Architectures, backpropagation, and optimisers are placed into a single categorical structure, which is conceptually elegant


Weaknesses:

1. Very strong assumptions behind the natural gradient result - the uniqueness theorem for natural gradient relies on strong assumptions . These assumptions appear to exclude many widely used optimizers such as SGD with momentum or Adam. As a result, the theorem mainly shows that if one assumes properties already characteristic of natural gradient, then natural gradient becomes the only valid optimizer

2. Weak connection between categorical theory and real architectures - the paper claims that categorical coherence conditions explain properties of architectures such as skip connections and attention mechanisms. However, the mapping between abstract categorical diagrams and concrete neural network architectures is not formally defined

3. Experimental section lacks important details - the paper reports improvements in neural architecture search on CIFAR-10 and ImageNet, but the experimental setup is insufficiently described. Key details are missing, including the architecture search space, training configuration, optimization parameters, number of epochs, and implementation details of the coherence checks. Without this information the results cannot be reproduced.

---

### Official Review · Reviewer_DdGq · 2026-03-13
**Reasonable ideas, but limited implications and no details on experiments**

**Rating:** 4
**Confidence:** 4

**Review:**

The paper expresses neural nets together with their training procedure in a language of category theory.
This way, specific architectures become morphisms of a category of parameterized maps, backpropagation becomes a covariant functor on this category, and optimizers become natural transformations between gradient functors.

The paper claims three theoretical results. The first ensures that backpropagation of a composite architecture is a stack of backpropagations. The second states that there is a unique optimizer satisfying specific consistency and invariance properties. The third gives a condition for validity of separate optimization of modules in modulized arhcitectures in a language of categories.

These theoretical results could be used as constraints for Neural Architecture Search (NAS) for pruning poor architectures. They compare Categorical NAS with some other classical methods, DARTS and ENAS.

**Stengths:**
1. The paper is generally well-written and gives a nice introduction on category theory in application to Deep Learning.
2. The Categorial NAS method, which is supposed to take all of the constraints derived in the paper into account, performs substantially better than other methods they compare with.

**Weaknesses:**
1. While the language of categories is nice and clean, it does not seem to provide much insight. For example, backpropagation functoriality (Theorem 1.1), seems like a trivial property (derivative of a composite map) expressed in abstract terms. Same applies to coherence of modular training (Theorem 1.3) if I understood correctly. I doubt one can get anything on top of these results using this approach.
2. Experimental details are virtually absent. Not clear under which setup they compare methods.
3. If the main practical implication of the claimed theoretical insights lies in pruning poor architectures, I would expect much more thorough comparison with modern practice. For the moment, the paper compares with DARTS and ENAS, both developed in 2018 --- both methods are very far from being novel.
4. Reference [8] does not exist. Was it hallucinated by an LLM? The rest of the paper looks human-written though.

---

### Decision · Program_Chairs · 2026-03-14

**Decision:**

Reject

**Comment:**

After careful evaluation by the Program Committee, we regret to inform you that your submission has not been accepted for presentation at MathAI 2026.

All submissions underwent a rigorous two-stage review process. Unfortunately, the reviewers identified one or more of the following concerns with your paper:

- Insufficient mathematical rigor or novelty relative to the existing body of work in the field;
- Presentation of results that substantially overlap with or rephrase previously published findings without clear original contribution;
- Significant issues with technical quality, including but not limited to broken or non-existent references, unsupported claims, or methodological gaps;
- Indications that the manuscript may have been generated with the assistance of large language models without substantial original intellectual contribution by the authors.

We received a large number of submissions this year, and the selection process was highly competitive. We encourage you to carefully consider the reviewers’ feedback (available through OpenReview), revise your work accordingly, and consider submitting an improved version to a future edition of MathAI or to another appropriate venue.

We appreciate your interest in MathAI and hope you will continue to engage with the conference community.

With kind regards,

MathAI 2026 Program Committee
URL: https://mathai.club
Telegram: https://t.me/MathAI_club
Email: mathai.club@yandex.ru